

# How you teach changes who you reach: understanding the effect of teaching modality on student engagement, content interest, and learning in undergraduate hydrology

Christine B. Georgakakos[1,2], James Knighton[1]

[1]Department of Natural Resources and the Environment, University of Connecticut, Storrs, CT, 06268, USA
[2] US Department of Agriculture, National Institute of Food and Agriculture

*Correspondence to*: Christine Georgakakos (christine.georgakakos@uconn.edu)

**Abstract.** There is a growing consensus that hydrology education should move towards student-led learning formats and simultaneously incorporate recent hydrologic technologies that reflect workforce expectations. There is a strong theoretical basis that supports an anticipation of improvements in learning outcomes from these shifts in teaching style; however, little empirical evidence has been collected to confirm this success. We measured the classroom impact of shifting between three teaching modalities: 1) instructor-led lectures, 2) student-led hydrologic modeling with the EPA Storm Water Management Model, and 3) student-led design evaluation studios of stormwater best management practices. Educational outcomes were measured with student surveys, direct observation of class activity, and student grades. In aggregate, the student population did not express a significant preference for one modality over another, yet individual students showed dramatic preferences for each modality. The total frequency of interactions between students and the instructor were similar across all three modalities; however, the frequency of student-initiated engagements (both total and unique engagements) significantly increased in both student-led modalities. Variations in student enthusiasm did not correlate with written assessment scores, possibly suggesting that alternating modalities improves interest in hydrologic science and increases perceptions of a positive classroom experience, without changing retention of hydrologic concepts. Our results suggest that multiple teaching modalities should be employed to engage the greatest number of students and generate enthusiasm for hydrology.

**Keywords**: hydrology education, pedagogy, student surveys, class time assessment tool, EPA SWMM, design evaluation





## 1. Introduction


Societal pressures related to flooding, drought, and residential, industrial, and agricultural water consumption are being exacerbated due to temporal changes in the global climate and spatial changes in human population density (Rama et al., 2022). A substantially greater number of people live under increased water-related risks (e.g., flood,

drought, decreased food security) now than in the recent past (Jongman et al., 2012, 2014; Smirnov et al., 2016). There is also broad evidence of declining forested ecosystem resilience that is likely attributable to changing hydrometeorological conditions (Forzieri et al., 2022; Seidl et al., 2017). These challenges necessitate a global workforce of engineers, scientists, and policymakers who are properly educated on both fundamental hydrological processes and the practical application of these concepts. A recent series of interviews with water resources

professionals indicated that graduates lacked critical workforce skills (Habib & Deshotel, 2018).  Expanding and strengthening the water resources workforce requires that we critically evaluate the capacity for teaching methods to both efficiently convey foundational ideas and inspire students to pursue future education, research, and work experience in hydrology.

There is broad evidence that student-led learning improves in-class experiences and retention of concepts in STEM education (de Jong, 2019). Research on STEM pedagogy has focused heavily on the effect of flipped classrooms model (where lecture material is introduced outside of the classroom, and class time is reserved for student-led exploration of the material) and suggest improved educational outcomes from student-led modalities (Berry et al., 2012). Hydrological education is evolving to include a greater focus on data-driven analyses, which provides ample

opportunities to introduce new learning techniques (Ruddell & Wagener, 2015). While informative, these conclusions of broader STEM educational research need to be validated in hydrologic education given the rapid changes in technology being introduced and the wide array of recent recommendations in STEM education.

New teaching modalities and technologies in the sciences have been proposed and implemented in educational

settings at a rapid pace; however, there has been relatively little critical evaluation of student outcomes (Coleman et al., 2019). Within hydrology, improved access to scripting languages (e.g., R, Matlab, Python) can possibly lower barriers for instructors and students to rapidly access large hydrological datasets and integrate these data into exercises (Lane et al., 2021; Slater et al., 2019). Access to high resolution hydrologic datasets will possibly support an improved focus on more relevant material, which could ready students for current water resources challenges

(King et al., 2012; Sabel et al., 2017). Simplified access to satellite-derived datasets can possibly allow classrooms
to leverage remote sensing to increase the spatial scale of problems (Maggioni et al., 2020; Wagemann et al., 2022).
Augmented- and virtual-reality visualizations of hydrologic model outputs have been proposed as educational tools
that could complement conceptual models of hydrological processes with fully graphical representations of water
fluxes (Tague & Frew, 2021; H. Xu et al., 2022; W.-W. Xu et al., 2022). Integration of computer simulation into
teaching exercises has long been explored for opportunities to increase student engagement above that of the
lecture-centric approach (Burt & Butcher, 1986). Several studies focused on student appraisals of in-class activities
suggest that course modules focused on hydrological simulations resulted in student perceptions of enriched
educational experiences (Gallagher et al., 2021; Knoben & Spieler, 2022; S. W. Lyon et al., 2013; Merck et al.,
2021; Pérez-Sánchez et al., 2022).


Despite prior positive student appraisals of updates to hydrologic curricula, other metrics of student success suggest
mixed results from shifting teaching modalities away from lecturing. Flipped classroom activities for water literacy
education found that in-class experiences were improved, but written assessments showed no significant
improvement (Moreno-Guerrero et al., 2020), a result mirrored in some broader STEM educational evaluations
(Akçayır & Akçayır, 2018; Chen et al., 2018; Troy Frensley et al., 2020). In another case, a subset of hydrology
students who were exposed to an interactive modeling platform performed slightly worse on a course assessment
than those deprived of access (Marshall et al., 2015). In contrast, a case study exposing students to the integration
of real-world data in a modeling project showed improved test scores following modeling activities (Sanchez et al.,
2016). It remains unclear if the near universal student reports of positive in-class experiences translate into
improved learning and workforce preparedness. New research is needed that explores the potential disconnection
between conclusions drawn from student perceptions of learning and student assessments.

Another challenge is that prior studies place strong emphasis on group mean outcomes of hydrology courses (i.e.,
average perceptions, average assessment scores), rather than the outcomes of individuals (Gallagher et al., 2021;
Knoben & Spieler, 2022; S. W. Lyon et al., 2013; Merck et al., 2021; Pérez-Sánchez et al., 2022). Student
preferences for information formats vary and any given teaching style may resonate most strongly with only a
subset of learners (Spoon & Schell, 1998). Studies that focus exclusively on mean outcomes may therefore only
reflect the majority learning preferences of the students present and lose critical detail on the number of individual
students reached with different teaching methods.




Our research contributes to the need to evaluate rapidly evolving learning techniques quantitatively and qualitatively in undergraduate hydrology. We evaluated the effect of introducing a sequence of three different teaching modalities (lecture, student-led hydrological modeling, and a student-led design evaluation) spanning topics in urban hydrology over a period of 8 weeks. Educational outcomes were measured with three parallel
methods: student surveys, continuous teaching observation of in-class engagements, and student assessment grades.

## 2. Methods

### 2.1 Course description

This study took place in a 3000-level undergraduate hydrology course at a state institution with about 25,000 undergraduate students. There were 25 students enrolled in the course, in years 3 and 4 in their degree programs. All surveys and experimental design was approved by the Institutional Review Board (IRB). The class consisted of three weekly meetings for 50 min each across 15 weeks. Prior iterations of this course followed a lecture style format across all 15 weeks. In this iteration, the final four weeks were modified to include: 1) traditional lecturing
(Lecture), 2) a hydrological modeling exercise (Model), and 3) a group-based water infrastructure design evaluation (Design). The second two modalities followed recent recommendations in that they were student-led (Thompson et al., 2012), incorporated real-world data (Sanchez et al., 2016), field measurements (Van Loon, 2019), and simulations with a hydrological model currently used in industry for water resources planning and design (Habib & Deshotel, 2018; Ruddell & Wagener, 2015). The instructor remained the same across all modalities and instructor
activities were recorded using a classroom time assessment tool (Fig. 1, Tables S1, S2).

The lecture section (three class periods, 150 minutes contact time) covered calculation of surface runoff from impervious surfaces (SCS Curve Number method and Rational method), introduction to the design of combined storm sewer systems, and an introduction to stormwater best management practice (BMP) design (BMP sizing,
loading ratios, infiltration rates). The lecture-oriented classes involved: approximately 15 minutes of the instructor reviewing administrative issues (homework due dates, scheduling review periods, clarifications on homework or reading), approximately 75 minutes of theory introduction on the white board, and 60 minutes reviewing practice problems (Fig. 1). The lecture included a homework assignment that reinforced the lecture material due one week after the lecture modality.






The interactive hydrologic modeling section (three class periods, 150 minutes contact time) tasked students with determining the approximate number of stormwater BMPs required to reduce stormwater runoff by 50% from a mixed urban-residential area with an EPA Storm Water Management Model (SWMM) (see supplemental material for the SWMM files used in the exercise). The first 50-minute period was dedicated to a walking tour of Best

Management Practice installations on campus to familiarize students with the infrastructure. In class, students organized into groups of two or three, familiarized themselves with the model, developed a new LID module, and used the model to simulate hydrological fluxes for a series of five historical storms. The classroom computer was used to project the SWMM model onto the main screen and was available for students to share techniques and improvements. Students were encouraged to iteratively modify the number, location, and dimensions of rain

gardens to design a stormwater management plan. Modeling results of all groups were reviewed and discussed together with the instructor. This modality involved: five minutes of administrative tasks, 15 minutes of introduction to Storm Water Management Model (SWMM) on the university campus, 110 minutes of group work on the modeling exercise, and 20 minutes of whole-class discussion. The instructor encouraged groups of students to continue working on the project outside of the classroom.


The design evaluation section (five class periods, 250 minutes contact time) tasked students with evaluating the hydrologic performance of nine existing stormwater BMPs on the university campus. Students were organized into groups of four or five and each group assigned a different BMP. Students were asked to do the following in the order of their choosing: collect field data using basic measurement tools (contributing drainage areas, BMP

dimensions, and soil properties), access hydrologic data from public repositories (NRCS Web Soil Survey, NCDC precipitation), perform a GIS analysis to confirm contributing area properties (slope, soil types), and perform desktop analysis using a runoff model (SCS Curve Number) and Excel mass-balance (bucket model) to estimate the return period for overflow of the target BMP. The instructor encouraged groups of students to continue working on the project outside of the classroom, culminating in the submission of a final report.


## 2.2 Data collection

All surveys and experimental design was approved by the Institutional Review Board (IRB) at the University of Connecticut. Student surveys were given at the end of each course modality focused on understanding their



preferences for each module (Table S3). Students generated a unique code that was written at the top of each survey. Student codes were linked to student names on an opt-in basis through the end of the course. In this manner, student survey responses were then linked with student grades. Surveys included both Likert-style and open-ended questions. Class time was allocated for survey completion.

Individual-level activity data was collected during each class of the three course modalities by an observer unaffiliated with the course. Data collected included instructor activities, student activities, questions originating from the instructor, and questions originating from students (Tables S1, S2). Question data was collected to allow for identification of the number of unique students engaged by the instructor and the number of unique students engaging the instructor per class. Observed student questions were not linked to grades or student surveys.


Finally, student performance on written assessments was recorded. We measured student understanding of the material with three short format written assessments given at the conclusion of each teaching modality (four questions each). We also collected grades from two long-format written assessments (15 questions each). One long-format exam was given prior to the start of the experiment (reflecting material covered in a lecture format for a

preceding 6 weeks), and the other at the conclusion (covering material from these 4 weeks). Grades were paired with the survey responses using unique student codes and de-identified for analysis.

## 2.3  Data analysis

We tested the hypotheses that 1) the total number of student-instructor engagements, 2) the total number of instructor-initiated engagements, and 3) the total number of student-initiated engagements varied between Lecture, Model, and Design modalities using a two-way ANOVA of regression slopes of question frequency versus cumulative time. With this and all subsequent hypothesis tests, we will discuss significance at the alpha levels of 0.1, 0.05, and 0.01. We then tested the hypotheses that the total number of individual students engaged increased

between Lecture and Model modalities, and the Lecture and Design modalities with a two-way ANOVA with "modality" as a variable.

Significant differences in the distributions of Likert scale responses were tested with non-parametric two-sample Kolmogorov-Smirnov tests between the Lecture versus Model and Lecture versus Design modalities. This



approach determines if the distribution of responses were significantly different, indicating variations in engagement across the entire class.

We tested if variations in student-reported preferences on the teaching modalities (i.e., Likert responses) were predictive of grade outcomes (i.e., did increased enthusiasm from a teaching modality result in changed grades).

We first computed the paired change in exam scores between an exam given prior to the three teaching modalities, and one given after the three modalities. We then grouped students based on each of the Likert responses, dividing the population into students who responded "strongly agree" and all other responses. We tested for significant differences in the distributions of grades between the clusters of students with two-sample Kolmogorov-Smirnov tests. All data analysis and visualization were completed in Matlab and R-Studio.


## 3. Results

### 3.1 Class activity distributions

Altering the teaching modalities shifted the activities of both the students and the instructor (Fig. 1). Under Lecture,
the instructor's role was primarily dedicated to communicating new content through the whiteboard and problem review, comprising approximately 75% of the class time (Fig. 1). Student-led activities (e.g., collaborating on practice problems; student chatter) comprised approximately 15% of class time. Under Model modality, the instructor role was similar, but dropped to approximately 20% of class time (Fig. 1). In both Model and Design, student-led activities increased to approximately 70% of class time, and included computer work, group work, and
individual or small-group engagements with the instructor (instructorwalktalk) (Fig. 2).

### 3.2 Student Survey results

Student surveys contained both Likert-style and open-ended questions. The Likert-style questions suggested that
there were not substantial differences between the student perceptions of the teaching modalities when averaged across all respondents (Fig. 3). For all modalities, a majority of students agreed that class time helped them to understand topics addressed (Fig. 3a), that their interest in hydrology increased during the lessons (Fig. 3b), and that they were interested in learning more about this particular topic (Fig. 3d). Only the design project excited a



majority of students about a career in hydrology (Fig. 3c). As group dynamics can impact student experiences, we
note that students reported largely positive group work experiences in both the modeling and design modalities.
(Fig. 3e). No significant differences in student preferences were observed across the three teaching modalities.

Though class-level responses suggested minimal differences between modalities, tracking individual students
through the three surveys made clear that individuals had clear preferences for some modalities over others. This
is demonstrated by students who responded either negatively or positively to the questions during one modality,
shifting toward the opposite responses in following modalities (Fig. 4).

### 3.3 Student in-class engagement

Across all modalities, the rate of all questions generated by the instructor, students, and both were approximately
linearly related to cumulative in-class time (Fig. 5). The frequency of instructor-initiated questions was significantly
higher in the Lecture modality than in Model (p-value = 0.001) and Design modalities (p-value = 0.000) (Fig. 5a).
The frequency of instructor-initiated questions was similar between the Model and Design modalities (p-value =
0.701) (Fig. 5a). Student-initiated questions were significantly more frequent in both the Design modality (p-value
= 0.000) and Model (p-value = 0.000) modalities than Lecture (Fig. 5b). Student initiated questions were also
significantly more frequent in the Design than the Model modality (p-value = 0.001). All modalities displayed
similar total engagements between the instructor and students (Lecture-Model p-value = 0.934; Lecture-Design p-
value = 0.734; Model-Design p-value = 0.550) (Fig. 5c).

The medians of unique student-initiated engagements (i.e., number of students who asked at least one question)
were 9, 13, and 14 for the Lecture, Model, and Design modalities, respectively (Fig. 6). Although the total number
of engagements were similar across all three modalities (Fig. 5c), the Lecture style format encouraged the fewest
students to engage (i.e., more repeat questions from a subset of students). The Design and Model modalities had
significantly more unique engagements than the Lecture modality (p-values = 0.000 and 0.002 respectively). The
Design modality showed significantly more unique students engaged than the Model modality at the α = 0.1
threshold (p-value = 0.081). Class modality was a significant predictor of student engagement in a two-way
ANOVA comparison at the α = 0.05 threshold (p-value = 0.028). There were no significant differences between
the number of unique students engaged by the instructor across the three modalities (p-value = 0.94) (Fig. 6).




## 3.4 Student Assessments

The mean short-format assessment grades of the Lecture, Model, and Design were 92.1%, 92.7%, and 91.5%, demonstrating strong student performance following each teaching modality. The median paired difference in
grades between the initial and final long-format assessments given to students showed a slight decrease in grades following these three teaching modalities of -2%. This change was not significantly different from 0 (p-value = 0.305) possibly due to the small sample size and large variance of changes in grades (standard deviation = 24.2%) (Fig. 7). Despite substantial differences in preference for teaching modalities (Fig 4), there were no significant differences in the distributions of long-format grade changes based on Likert responses (Fig. 7).


## 3.5 Qualitative Results from Student Questionnaires

After completion of the lecture modality, students expressed the least interest and most concern about the
mathematical calculations required. Students felt unprepared by pervious courses and uncertain if they would be prepared for careers in this field (5 students mentioned this concern).  Student stated " I think it may be difficult to incorporate hydrology into my career without learning more advanced math" (student 15). Students were excited by utilizing online tools (4 students) and drawing connections between theory and practice (4 students). Five students remarked that the modality had increased their excitement about careers in hydrology, especially in the
connection of concepts: "It really just made me more eager to get into a water based career. Seeing all these concepts come together was very cool!" (student 9).

After the Model modality, which included a walking lecture of green infrastructure, 10 students mentioned the infrastructure development concepts and tour as the most interesting component for them, with an additional 3
mentioning the software applications of new concepts.  Seven students mentioned the logistical difficulties of learning new software. Four students mentioned this modality as increasing their interest in a career in hydrology, with one student even mentioning that "it seems less intimidating" (student 2) while another noted "the modeling helped me realize a new resource that I can use in the future" (student 4).



After the design modality, 5 students each mentioned data collection and field work as the most interesting components of the modality, while 6 students mentioned the applicability of their new knowledge as the most interesting part. Four students mentioned the modeling as the least interesting component of the design analysis (primarily mentioned GIS watershed delineation and runoff modeling). Two students mentioned the decision making that goes into some modeling, "how uncertain the choices a hydrologist must make [are]" (student 16).

## 4. Discussion

### 4.1 Teaching Modalities and Educational Outcomes

Prior studies have posited that student-led activities will improve educational outcomes in hydrology (Thompson et al., 2012). Our study attempted to validate this hypothesis by measuring learning outcomes from three teaching
styles: lecturing, student-led hydrological modeling, and student-led design evaluations of stormwater installations. Our experimental design heeded the conclusions of recent studies in that we included real-world data (Sanchez et al., 2016), field techniques (Van Loon, 2019), and hydrologic software with career relevance (Habib & Deshotel, 2018; Ruddell & Wagener, 2015). Our shift from instructor-led to student-led modalities resulted in the anticipated shifts in classroom focus from the instructor to the students (Fig. 2) without a change in the number of unique
students engaged by the instructor (Fig. 6), suggesting a successful implementation.

Prior hydrologic education investigations studying the impact of shifting teaching modalities away from lecturing have focused heavily on aggregated student perceptions of learning (Gallagher et al., 2021; Knoben & Spieler, 2022; S. W. Lyon et al., 2013; Merck et al., 2021; Pérez-Sánchez et al., 2022). The result that we observed was
similar to prior studies noting improved student perceptions of learning as well as foundational research on student-led education. Student-led exercises enriched classroom experiences with an increased frequency of student-initiated engagements (Fig. 5b), more individual students engaged during the class period (Figs. 3 & 6), and students reporting slightly more positive attitudes towards the field of hydrology (Fig. 2). Perhaps most critically, Lecture, Model, and Design each appealed to different subsets of students (Fig. 4). By covering a broader range of
teaching styles, a greater proportion of students were engaged for at least one of the modalities, which may have increased the number of students who considered continuing on to hydrology related careers (Fig. 4c). This



important result is obscured when looking only at mean survey responses (Fig. 3), suggesting studies should track shifts in student-level engagement.

Our findings also showed a dissonance between student reports and written assessments that has been highlighted in several prior studies (Fig. 7) (Akçayır & Akçayır, 2018; Chen et al., 2018; Moreno-Guerrero et al., 2020; Troy Frensley et al., 2020). Recent research on the application of student-led learning in science curricula suggests improved classroom experiences can possibly be negated by out of classroom experiences (Akçayır & Akçayır, 2018; Chen et al., 2018). The student assessment scores that we observed did not indicate that altering the teaching
modality had a lasting impact on the retention of hydrology concepts (Fig. 7). One possible explanation for these results is that students were strongly engaged during the entire 4-week experiment. This explanation is supported by the mean short-format assessment grades (all means above 90%). It was statistically unlikely that students would show an improvement in grades that were already very close to the upper grade threshold.  Another possibility is that student evaluations in the form of written exams may fail to adequately measure student achievement (Steve
W. Lyon & Teutschbein, 2011). Alternately, a decreased emphasis on individual assignments (and increased emphasis on group work) outside of the classroom possibly lowered student engagement, offsetting the educational gains made in the classroom. This explanation may be less likely as there is some evidence that collaborative experiences outside of the classroom improve student learning (Bosman & Fernhaber, 2019). Finally, teaching modality may not impact the mean long-term retention of hydrology concepts across all students. If this is indeed
true, employing varied modalities may still yield a net positive effect for hydrologic education, as increased student enthusiasm is possibly just as critical as memorization of concepts to long-term success and workforce health. Given the differences in conclusions that have been drawn from varied metrics of classroom success, we propose that it is critical that hydrology education research continue to employ methods to quantify student outcomes beyond student perceptions.


## 4.2 The Role of Educational Research in Decentralized Hydrology Curricula

Hydrology curricula are relatively decentralized, where most college-level educators use a wide variety of texts and often custom course materials which presents some advantages (Merwade & Ruddell, 2012; Ruddell &
Wagener, 2015; Thompson et al., 2012; T. Wagener et al., 2012; Thorsten Wagener et al., 2007). Heterogeneity in hydrologic curricula suggests that the field is well positioned to adapt to relevant case studies highlighting local





and regional water issues. A decentralized hydrology curricula can also be rapidly updated to meet changing social or environmental conditions, advances in the field, or changing technologies (Merwade & Ruddell, 2012). New challenges at the human-water interface can possibly be integrated into undergraduate curricula to keep courses
relevant to the immediate needs of society and the interests of students (King et al., 2012; Sabel et al., 2017). There is evidence that framing hydrological education in the context of real-world data and scenarios enhances learning (Sanchez et al., 2016). Qualitative evaluation of case studies of combined instructor- and student-centered teaching methods involving project-based learning, case-based learning, and discovery learning led to the hypothesis that these approaches improve educational outcomes (Thompson et al., 2012).


We observed clear gains in student engagement by varying course modalities (Figs. 4, 5, 6), but not necessarily a clear improvement in mean long-term retention of concepts (Fig. 7). Evaluating this result in the context of prior research on hydrology education is challenging because few studies have employed multiple methods of quantifying student outcomes beyond questionnaires distributed to students (Gallagher et al., 2021; Knoben &
Spieler, 2022; S. W. Lyon et al., 2013; Merck et al., 2021; Pérez-Sánchez et al., 2022). Proposed improvements to teaching are likely outpacing our capacity to critically measure their value. Centralizing hydrology with standard teaching styles and materials could help to ensure a critical evaluation of all new proposed modifications; however, the positive benefits of a decentralized hydrology curricula likely outweigh the benefits of centralizing teaching styles and material. We therefore propose that the advancement of a field reliant on decentralized curricula should
strive to adopt centralized educational evaluation criteria to support the continued advancement of learning.

**4.3 Practical Recommendations for Student-Led Modalities**

Our research yielded several practical recommendations that could possibly improve the implementation of altered
teaching modalities which we evaluated. This section summarizes those recommendations.

Specific considerations for hydrologic modeling software could lower barriers for student learning. Some modeling software does not work on all operating systems. Our hydrologic modeling modality allowed students to use their personal laptops. US EPA SWMM, which we employed in this exercise, is currently offered only as a compiled
executable for PC (source code is also available). This complication resulted in a delayed start for several students and necessitated that they partner with other students. The use of a controlled computer laboratory that standardizes



hardware and software would limit the complexity of the assignment and allow for an increased focus on the intended learning objectives.

Student-led learning can possibly benefit from instructor interventions in specific circumstances. Throughout the Model and Design modalities, several students were absent when groups were formed. These students were placed into groups upon returning to the classroom. Group dynamics formed rapidly, and the students joining late may have had less positive group experiences as a result. Instructor intervention, such as connecting the absent student to their group via email, may have improved group experiences. Similarly, several collaborations across groups

occurred organically in the Model and Design modalities when all members of one group were unsure on how to proceed. These collaborations appeared to benefit the learning of all involved. Such collaborations could be actively encouraged through planned instructor interventions such as a peer-review, student critiques, and online discussion to allow for further collaboration outside of the classroom (Bosman & Fernhaber, 2019).

Our experimental design consisted of each modality lasting for a minimum of three in-class periods. This choice was intentional to allow students to become fully immersed within each teaching style and to allow for more ambitious activities (e.g., development of a hydrologic model, collection of hydrologic data from a field site). The length that we employed possibly limited the total number of classroom teaching variations. The outcomes of introducing a wider variety of smaller activities on a daily scale on student learning and experiences should be

evaluated.

## 5. Conclusions

We modified the teaching modality in a moderately sized undergraduate hydrology course to cover the following

teaching styles over a 4-week period: instructor-led lectures, student-led hydrologic modeling exercise, and a student-led design evaluation studio. Across the entire student population, there were not clear preferences for any one teaching modality; however, individual students showed strong preferences. The modeling and design modalities both encouraged a greater number of student-initiated engagements with the instructor (both total and unique engagements) in comparison with the lecture format, suggesting improved student interactions. Written

assessments did not demonstrate significant differences in learning across the three teaching modalities. Our research contributes to a growing body of evidence that highlights that different conclusions might be drawn from

different metrics of student outcomes in hydrologic education. Monitoring in-class engagement, student perceptions, and learning outcomes should be expanded to changes to pedagogical approaches to understand impact and benefit on students. Our research broadly indicates that a higher diversity of teaching methods benefits the

learning environment.

## 6. Acknowledgements

This work is supported by AFRI Education and Workforce Development Program [grant no. 2021-67034-
35182/project accession no. 1026507] from the USDA National Institute of Food and Agriculture. We would like to thank the students who participated in this study for their participation and insights.

## 7. Code/Data Availability

All        code        and        data        are        publicly        available        on        Hydroshare        (
https://www.hydroshare.org/resource/1fcbb7cc3d2b4a80946601c626868349/).

## 8. Author Contribution

The authors contributed equally from funding acquisition to project administration, data analysis,  and manuscript writing and editing.

## 9. Competing Interests

The authors declare no competing interests.

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

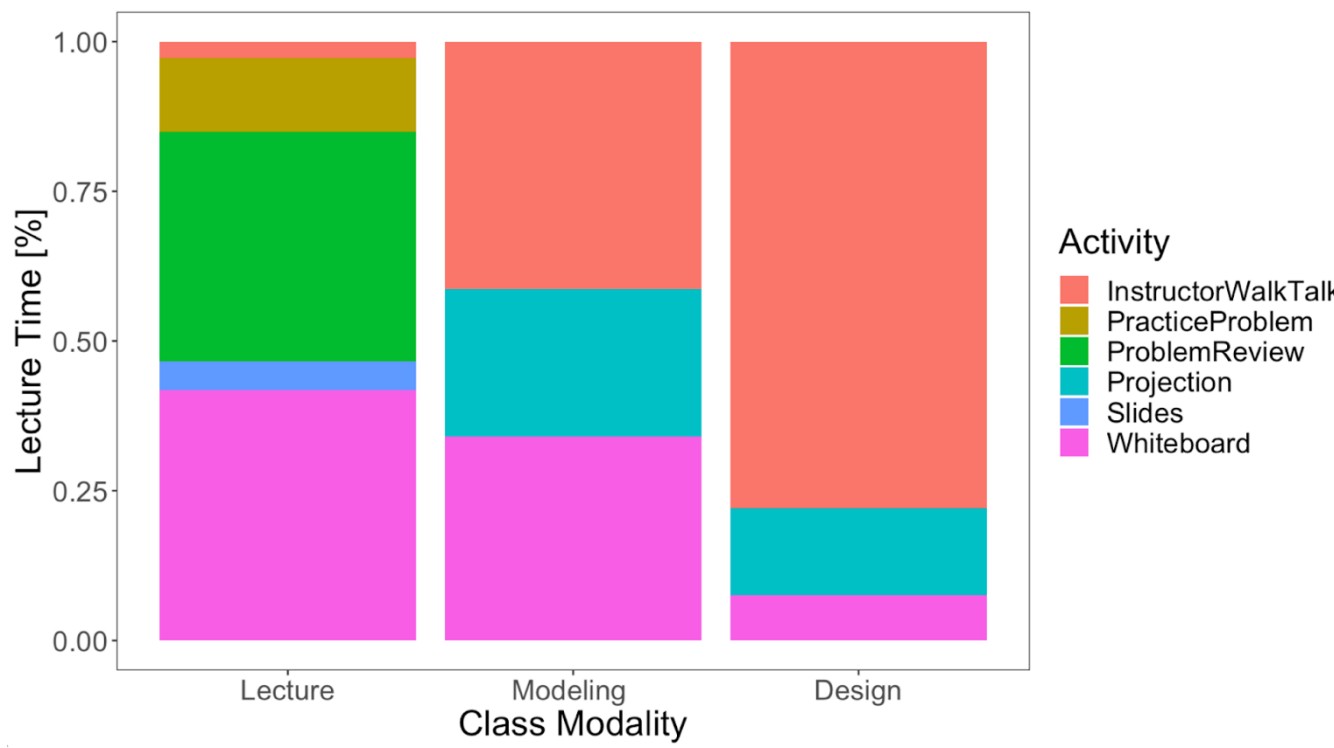


**Figure 1: Distribution of instructor activities directed toward the class as a whole for each modality.**



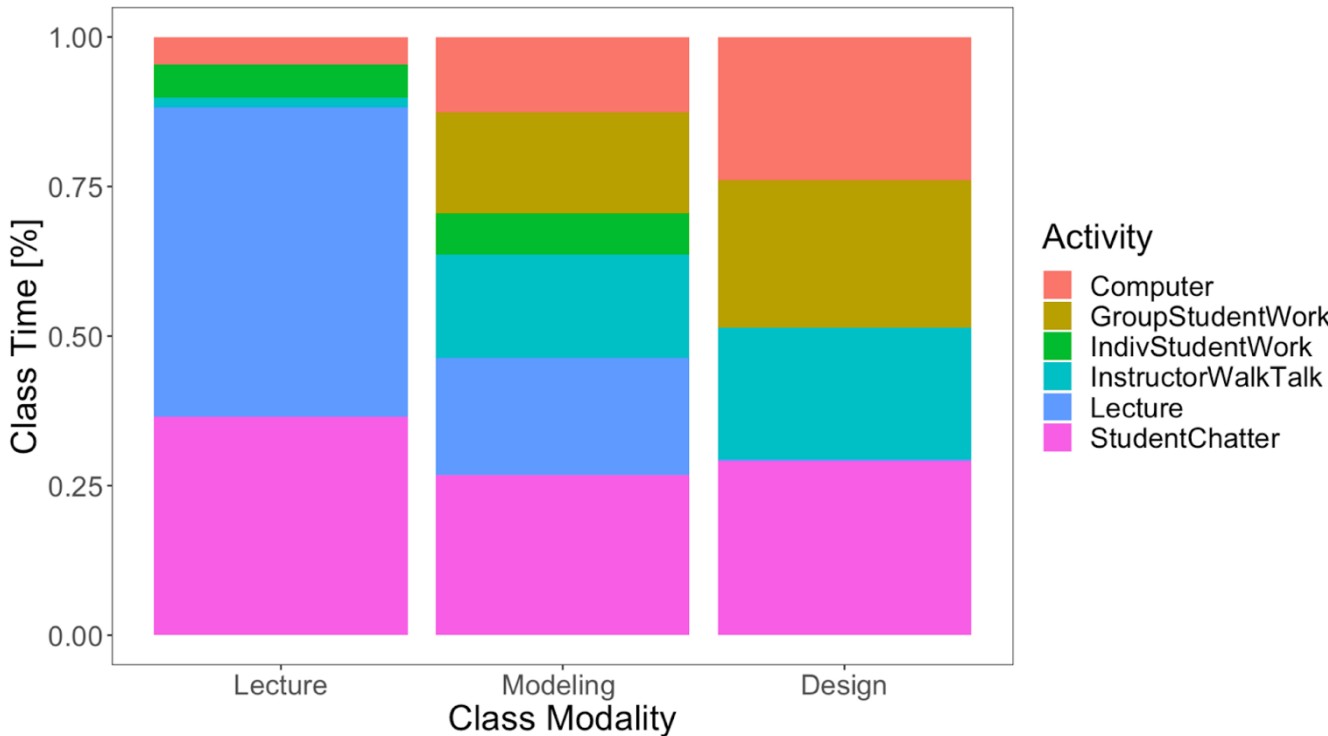

**Figure 2: Distribution of student activities across the three course modalities**





**Figure 3: Student survey Likert responses to career, content interest, group dynamic, and understanding questions.**



**Figure 4: Alluvial flow diagram tracking individual student responses between surveys. Students responded**
**to statements with strongly agree (SA), agree (A), neither agree nor disagree (N), disagree (D), or strongly**
**disagree (SD).**





**Figure 5: Cumulative student engagements over the course of a class in each teaching modality (Lecture, Model, Design). Questions originating from the instructor (A), from students (B), and total combined questions (C) are plotted separately. A comparison of slopes of the linear regressions were completed using an ANOVA, with p-values comparing differences between each of the three lines listed on the plots.**




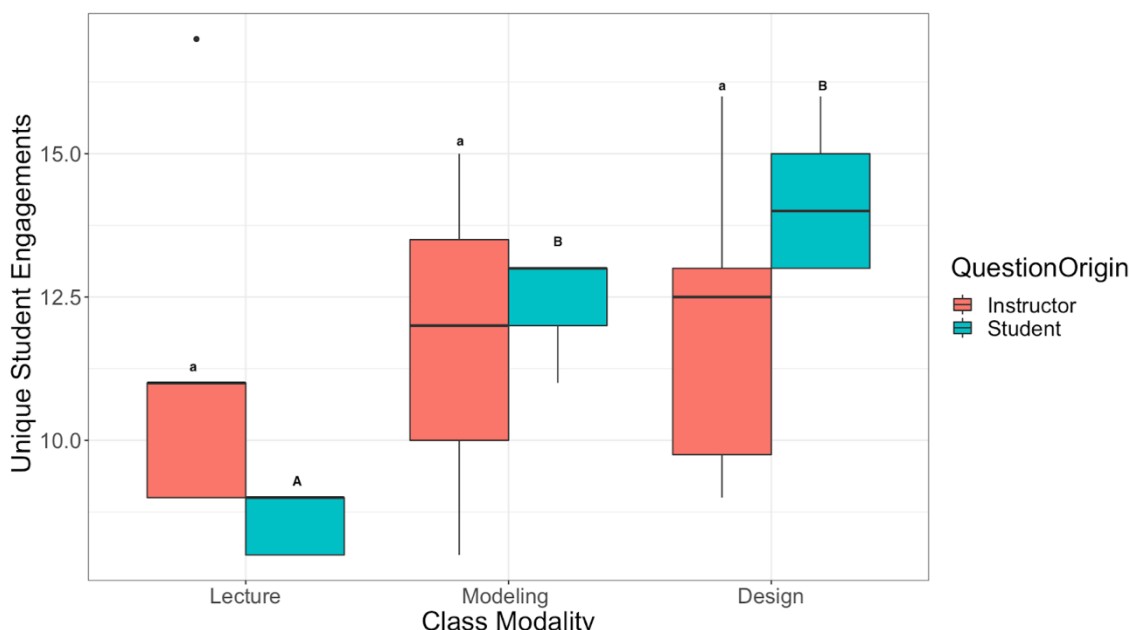

**Figure 6: Unique engagements originating from students (blue) and the instructor (red). Statistical difference is noted by capital and lowercase letters for students and instructors originating questions respectively.**




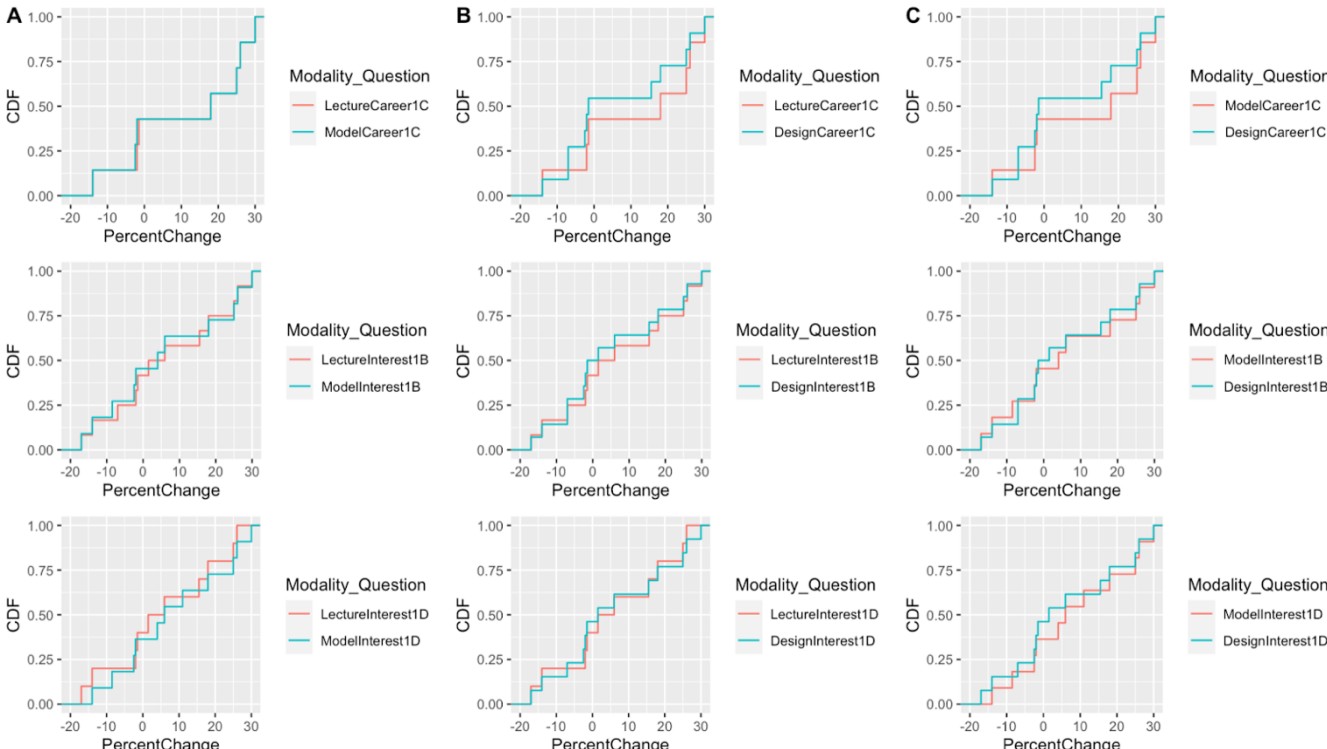

**Figure 7: Cumulative distribution functions of student grades for students who responded to the career and interest questions with a '4' or '5' on the Likert scale of '1'-'5'. '1C', '1B', and '1D' refer to the Likert survey questions to which each response corresponds. Column A compares Lecture and Model modalities, column B compares Lecture and Design modalities, and column C compares Model and Design Modalitites.**