# Peer review of "How you teach changes who you reach: understanding the effect of teaching modality on student engagement, content interest, and learning in undergraduate hydrology"

_EGUsphere, 2022_

## Author Comment (AC1)

**Responses to the Comments of Reviewer 1**

**Summary: In my opinion this is a thoroughly documented experiment and a well-written paper. The main message that different teaching styles appeal to different students is perhaps not entirely unexpected, but it might still be helpful to those with teaching responsibilities to see that this holds true within a hydrological teaching context.**

1. **I do think that a few things can be clarified (see comments in the attached .pdf) and that section 4.2 and 4.3 in the Discussion could use some work. One element that could be added here is a brief discussion of the authors' experience of the benefits and drawbacks of creating materials for these various teaching styles, and touch on the question "is it worth my limited time?" that readers of this paper may be asking themselves.**

We thank the reviewer for their comments and suggestions. We understand the reviewer's point and will include a revised discussion. Please see our response to comment 28 from this reviewer for a detailed discussion related to "is it worth my limited time". We have also modified the title of discussion section 4.3 to be more inclusive of these themes. Please also see our response to comment 24 from this reviewer addressing the changes to section 4.2.

*We respond to each individual comment below.*

2. **Line 51: Is there a reason to assume that hydrologic education is somehow different from the broader STEM fields, that leads to this need?**

We will rewrite this section as follows:

"Hydrology possibly differs from broader STEM fields in several ways. First, geophysical challenges vary regionally, often resulting in region-specific educational case studies (e.g., flooding, drought, urbanization, agriculture runoff). STEM fields outside of the geosciences (e.g., medical research, chemistry) may have less regional variation in case studies. Second, compared to other STEM fields, hydrology curricula are decentralized. There have been reasoned calls to both decrease (Ruddell & Wagener, 2015) and increase the variation in hydrology curricula (Sanchez et al., 2016) concurrent with recommendations for new educational tools (Merck et al., 2021; Pérez-Sánchez et al., 2022; Lane et al., 2021; Slater et al., 2019). Given the regional nature of the geosciences, as well as variations in recent recommendations for future educational directions, there is a need to validate broader STEM findings within hydrology."

3. **Line 59: This is ambiguous. Do the authors mean an improved focus "on more material that is relevant [like high-res datasets are]" or "on material that is more relevant [than high-res datasets are]"?**

Thank you for identifying this phrase as ambiguous. We will change the language to more clearly reflect the "career-relevance" of using teaching tools such as high-resolution datasets.

4. **Line 87-89: I agree with this statement, but the authors may need to consider that studies investigation student perception of (hydrology) courses may be limited in the kind of assessments they can employ. Any assessment involving student grades can by definition be done at the level of the individual, but assessment involving student perceptions may need to be done through voluntary student participation in custom questionnaires. Teachers may be restricted by institutional policy in what they are allowed to measure through such informal assessment methods: policy may dictate that learning assessments must be performed at an aggregate level to guarantee anonymity for the students. Equally, students may be unwilling to participate in voluntary assessments at the individual level, if the perception exists that their answers may easily be traced back to themselves.**

We acknowledge that this study required substantial effort and institutional approval through our Institutional Review Board (IRB) to collect three different forms of assessments and de-identify students prior to analysis. This effort is likely infeasible to deploy in every class. We did not intend to suggest that these methods come at no expense. Instead, we meant that the results of prior studies that focused only on group mean perceptions may be missing detail and that this necessitates new studies that investigate this possibility. We will clarify this in the revised text as follows:

"Studies that focus exclusively on mean outcomes, often performed out of necessity to protect student privacy and ensure anonymity, may therefore only reflect the majority learning preferences of the students present and lose critical detail on the number of individual students reached with different teaching methods. Though anonymizing individual-level data to ensure student privacy may require additional personnel, the added data provides critical detail for assessing teaching modification impacts."

5. **Line 100: This terminology may be unfamiliar to readers outside of the US. Rephrasing this in terms of number of preceding years of hydrology education and expected knowledge/skill level will be more helpful to an international audience.**

We will change the terminology as recommended to be an "upper-level undergraduate course". We will also include expected courses taken prior to enrollment in this course.

6. **Line 110: If the following sections only discuss the final 4 weeks that were modified (as they appear to be), than it would be good to say so.**

We will restate that this modification happened in the final four weeks of the course as follows:

"The instructor remained the same across unmodified sections and all three modalities in the final four weeks of the course…"

7. **Line 114: The remainder of the text seems to use BMP as a noun to describe management infrastructure (i.e. things, instead of practices). Is this common use of the phrase?**

The term "BMP" to describe a stormwater best management practice installation is common practice and replaceable with "Green Infrastructure (GI)" and "Low Impact Development (LID)". "BMP" is the most commonly used abbreviation across academic fields, which is the reason we chose it. Unless it is recommended otherwise, we will continue to use BMP throughout the manuscript for consistency. And will ensure that all references use the same terminology.

8. **Line 131: Line 124 seems to suggest that this took 50 minutes.**

Thank you for identifying this error. The walking tour, though introducing specific stormwater management practices further studied in both the modeling and design sections, is considered a lecture as the students are primarily passively receiving information. This was the way we initially split the data for analysis, and the methods will be modified to reflect it. The total class times in each of the lecture, modeling, and design modalities will be adjusted to reflect the data. The tour did last for 50 min and the SWMM introduction was 15 min. We will clarify which time frames mentioned were total modality time versus an individual class period to reduce confusion.

9. **Line 141-143: I understand that these practices are part of designing infrastructure, but they seem to overlap in scope with the class periods dedicated to modeling. How can student preference for either modeling or design exercises be cleanly assessed if modeling is part of both?**

The "model" in the design phase of the class was a simple spreadsheet water balance (ΔStorage = Runoff - Infiltration - Overflow). Students were allowed to calculate runoff and infiltration with any method that had been introduced previously at any point during the course. It was misleading to refer to these calculations as a hydrologic model. We will clarify this in the revised text that these were spreadsheet calculations that did not differ substantially to the hand calculations that students were familiar with to this point in the class. Spreadsheets were used only to save time, not to facilitate the simulation of a more complex series of interrelated processes.

The Model phase of the class was carried out using a coupled H&H model simulating many physical processes in parallel (i.e., snowmelt, evapotranspiration, groundwater flow, runoff and stormwater detention in distributed Low Impact Development). Tasks centered on building-, compiling data for-, running-, and analyzing the results from-the model were substantially different from what the students had encountered previously.

We will clarify the differences between the two computer-oriented tasks in the revised text.

**10. Line 149: Were students given this survey before or after their grades for the modality were made known to them?**

All surveys were given at the end of the final class period before grades were known. We will clarify this in the revised text.

**11. Line 149: *Insert* 'And'**

This was a typo that we will resolve in the revision.

**12. Line 151-152: How many out of the 25 students opted-in on linking their surveys to grades?**

The class had 25 total students. One student had to leave the course for personal reasons shortly after the midterm. A total of 20 students agreed to participate. The response rate was 20/24 = 83.3%. We will clarify this in the revised text.

**13. Why do the percentages of InstructorWalkTalk not match between Figures 1 and 2? They appear to me to describe the same activity.**

InstructorWalkTalk in the two figures described the same activity, but from two different perspectives: Figure 1 from the perspective of the instructor and Figure 2 from the perspective of the students. If we consider just the Design Module, while the instructor may spend most of the time walking around the classroom and engaging with students, the students in contrast are doing additional activities while the instructor cycles in this manner (e.g. working on a computer or in small groups). To clarify, we will remove InstructorWalkTalk from the student perspective figure to reduce co-occurring activities. Thank you for highlighting this confusing point and allowing us to clarify it. We will also change the color format of both figures to clarify that repeated colors do not reference the same activity. Instead we will use monochromatic color scales to increase accessibility of the figures as shown below in Figure 2:

[Figure]

**14. How many students filled out each of these surveys?**

Out of the 20 that agreed to participate in the research, the lecture survey was completed by 18 students, the model survey by 16, and design by 18 due to absences. We will clarify this in the revised text.

**15. This way of visualizing student preference seems to lose information, because the preference of any individual student cannot be tracked across the three teaching modes. Separating comparative preference between Lecture and Model on the one hand, and Model and Design on the other hand means it is unknown how student perception of Lecture compares to their perception of Design. A possible way to connect responses from an individual across all three teaching modalities could be a Parallel Coordinate Plot, with the three course types on X, student response on Y.**

Per the reviewer's suggestion, we tried a parallel axis plot and it masks more information than it shows. The figure below is the result of the first question on Fig 4. This graph occurs because the data are all discrete so many lines overlap. The alluvial diagrams, while imperfect, show the proportions of students.

[Figure]

We have modified the alluvial diagram to contain another column showing lecture-design variations. We note that the total number of reported outcomes varies slightly as the alluvial diagram will only show results when individual students completed both surveys. Two students did not complete the lecture survey, four did not complete the model survey, and two did not complete the design survey. We will include this figure in the revised manuscript.

[Figure]

16. **Assuming each dot represents an observation of cumulative number of questions asked, then it's unclear to me why these plots do not show a cumulative progression**

**of number of questions asked, but rather bounce up and down around a generally upward trend.**

**See for example Fig. 5a, red dots:**

**- No questions for the first 15 minutes**

**- 10 questions asked at minute 19**

**- < 10 questions asked between minutes 30 and 37 or so**

**- A jump to 30+ questions asked around minutes 38, 39**

**- Back to 10 questions asked at minute 40**

**- etc.**

**I expect I'm missing something here but it's not clear to me what.**

These figures show the overlapping cumulative questions across multiple class periods. We will replace the data for each class period, previously represented by points with lines distinct for each class period so that the cumulative questions are presented more clearly and the variability between classes clearer. The proposed modification is below. We will also clarify this in the the figure caption.

[Figure]

**17. Line 231-233: Is there a record of the types of questions asked? Having run tutorials myself I can imagine that at least a number of the questions asked during the modeling and design courses were along the lines of "when is the assignment deadline?" and "where can I find the assignment data?" Were such questions filtered out so that the graph shows only questions related to understanding/working with the presented material?**

We did not record questions asked by students live in class, but we can confirm that they were a mix of high level questions about hydrological processes as well as more routine questions. We considered any engagements to be worth counting, even if questions were only to clarify something related to an assignment. Therefore, the engagements were not filtered based on content. This detail will be added to the methods. The result shown on Figure 6 demonstrates that we were getting only 15/25 students engaged during the design module, which engaged the most individual students. We speculate that if students are left wondering about a particular detail of an assignment, they may not be fully capable of absorbing the more detailed hydrology concepts that are being presented. We also find (qualitatively) that once a student breaks the ice by asking a question out loud, they become more likely to ask subsequent questions in class, which may be of a more detailed nature.

**18. Similarly, group work tends to be noisy compared to class lectures. Where one question and one answer suffices to convey that information to all students present in a lecture scenario, the same question may need to be repeatedly answered if asked by separate groups in a group work scenario. Where such duplicates filtered out?**

Duplicate questions were not filtered out because we were interested in the total unique engagements, with each unique student counting as an engagement. However, upon receipt of duplicate questions the instructor addressed the class as a whole or wrote relevant information on the board, reducing further duplicates.

**19. To my understanding:**

**- The lecture focused on surface runoff, storm sewer design and storm water best management practices.**

**- The model exercise focused on a field trip outside, optimization of infrastructure placement, model building and the running of simulations.**

**- The design project focused on evaluating the performance of infrastructure on campus, data collection, data access, GIS analysis and basic modeling.**

**Between the three modalities, not only the teaching style changed, but also the topics the student were tasked to work on. Therefore I think this statement as is, is not supported by the result of this experiment. Unless the topics can be considered**

**irrelevant for student engagement (which I don't think can be said), I believe this statement should read:**

**"By covering a broader range of topics and teaching styles, ..."**

The reviewer's interpretation is correct. With respect to the rest of the course, these three modalities covered the same general topic: stormwater management. We do agree with the point the reviewer is making and will adopt the recommended wording in the revised text.

**20. Line 303: The reader might benefit from having a short list of examples of what such out-of-classroom experiences can be.**

Thank you for highlighting this point of interest. The 'experiences' to which we are referring are out-of-classroom pre-lecture preparations by the students that enable, if completed, maximum in-person learning. We will clarify this in the text as follows:

"Recent research on the application of student-led learning in science curricula suggests improved classroom experiences can possibly be negated by lack of out-of-classroom preparation by students (Akçayır & Akçayır, 2018; Chen et al., 2018)."

**21. Line 304-305: If I understand the methodology correctly, this is based on student performance on three written assessments (i.e. tests), taken after each of the three modalities concluded. Lectures transfer knowledge, but modeling exercises and design projects transfer additional skills (data handling, debugging, handling group dynamics, etc). Did these written tests account for transferable skills gained during the three different teaching styles? Students may have acquired extra skills that are not evident in Fig. 7.**

The longer-format exams were focused on knowledge. The three short format assessments included questions that accounted for an understanding of model functionality and required data manipulation. The assessments were designed with the goal of testing the most relevant skills of each section. We will make the assessments available as part of the supplemental material.

**22. Line 323: Relative compared to what other fields?**

Thank you for identifying the lack of clarity of this phrase. We intended to reference the cited study, which found most hydrology educators teach from independent material, with fewer than 20% using community-generated education materials, which presents both advantages and disadvantages. We will modify the phrasing so that rather than comparing hydrology to other fields, we focus on variation within the field.

**23. Line 325: Referencing should be consistent. These two publications are by the same person.**

Thank you for catching this citation error. It will be corrected and all other references checked.

**24. I don't think Section 4.2 in its current shape adds a lot to this paper. The listed benefits of decentralized curricula seem debatable to me (why would a 1000 different hydrology professors scattered across the world be able to update their teaching materials more quickly when they all work individually?) and the suggestion for centralized education evaluation criteria comes out of left field.**

Thank you for identifying this discussion section as unclear. We will further explain the advantages of a decentralized curriculum to clarify its inclusion in this manuscript. And will include the questions outlined in comment 25 as questions that will need to be addressed if we are to centralize the curriculum. We will modify the explanation of advantages of decentralized curriculum as follows:

"Heterogeneity in hydrologic curricula suggests that the field is well positioned to adapt to relevant case studies highlighting local and regional water issues, which may increase student interest in the field by grounding examples in familiar locations. Similarly, a decentralized hydrology curricula can also be rapidly updated to meet changing social or environmental conditions, advances in the field, or changing technologies by avoiding bureaucratic delays in implementing changes (Merwade & Ruddell, 2012)."

**25. Line 344-345: If this highlighted sentence is the main point of this section than I think the authors should expand on it. Key questions that this statement invites are:**

**- Who in the community or how should the community decide what these evaluation criteria are?**

**- What aspects of education should these criteria cover?**

**- Why does the hydrologic community need to invest effort into this instead of relying on findings from the broader STEM research into effective education?**

**I'm not convinced that these questions can be easily answered.**

More than proposing a complete centralization of the curriculum, we acknowledge that a centralized curriculum would allow for widespread use the pedagogies that have critically evaluated. A more useful tool would be a set of practices tested in the hydrological sciences that courses could be built from with regional and local examples, retaining the flexibility of a decentralized curriculum with the robustness of a centralized methodologies. We will rephase the sentence highlighted here to include this perspective, and include the questions outlined above as necessary considerations for forming a centralized hydrology-focused pedagogical best practices list. We will revise the highlighted section as follows:

"We therefore propose that the advancement of a field reliant on decentralized curricula should strive to adopt a set of centralized, and evaluated pedagogical best practices that can be applied across a diverse curriculum, retaining both the flexibility of a decentralized curriculum and allowing implementation of tested teaching practices.  Establishment of such a set of practices requires consideration of who is included in this teaching and learning community, practices to be evaluated, and valuation within the field of hydrology and academia at large of time spent on evaluating teaching practices."

> **26. Line 350: Recommendations 1 (make sure the right computational infrastructure is ready to go) and 2 (take care when forming student groups when a number of students are absent) seem good recommendations to me, but also quite general (and possibly not very surprising) ones. Recommendation 3 relates to the way this experiment was designed and not to Student-Led Modalities, which does not fit the title of this section. The authors might consider splitting this section in two or rephrasing the current title.**

Thank you for citing this inconsistency.  We will modify the title of this section to be "Practical Recommendations", and also include a discussion of pros and cons as suggested and outlined below in our response to comment 28.

> **27. Line 353-355: This seems contradictory to me, or did none of the students work on Apple machines? Also, although I realize that these terms are often used interchangeably colloquially, "PC" (personal computer) should probably be replaced by "Windows" (the specific operating system that imposes certain conditions on the compiled executable). [follow-up] thinking about this further, is the intended meaning of these sentences something like the following?**
>
> **"Our hydrologic modeling modality did not specify any requirements for student laptops. However, US EPA SWMM, which we employed in this exercise is currently only offered as a compiled executable for Windows (though source code is also available). This led to a complication for students without access to a laptop with Windows, or the knowledge to readily compile source code themselves, resulting in a delayed start ..."**

Thank you for identifying this terminology error.  We will clarify this in the text as recommended and correct the use of "pc." Three students did use Mac laptops. Two needed to pair with other students to complete the exercise. One successfully ran EPA SWMM on a Windows emulator, but this required substantial effort for both the student and instructor.

> **28. Line 389-390: This recommendation might be coupled to a discussion item, where the authors outline the practical implications of using a higher diversity of teaching methods. Teaching is only part of most academics' workloads, and prepping and teaching the same material in three different ways takes longer than doing so in only**

**one way. Some discussion of the (authors' experiences of the) pros and cons of offering multiple teaching styles could be a very helpful addition to the paper.**

We will modify the title of section 4.3 to be inclusive of the discussion recommended here to "Practical Recommendations", and will include the following discussion to highlight the points recommended:

"We recognize that modifying existing curricula to include a wider variety of teaching modalities may be initially time intensive for instructors, and formally evaluating those changes additionally intensive. However, along with others (e.g. Wagener et al., 2007), we recommend an expansion of assessing pedagogical outcomes in hydrology to ensure implementation of best practices."

"And lastly, a data-rich experimental design such as the one discussed in this experiment with student engagement and participation data, survey responses, and de-identified grade analysis, may be infeasible for a single instructor to accurately collect alone in a traditional classroom setting. Classroom data collection by a third party enabled the depth of data collection required for this analysis and allowed the instructor to focus on preparation of teaching materials. Collaborators in university teaching and learning centers may already be trained on collecting this type of data and could support research to reduce time required of the instructor. Although the time required to (1) generate novel teaching materials and (2) evaluate novel practices is non-negligible, we believe the impact on students and ability to make data-driven decisions is meaningful."

**29. Using red for agreement feels intuitively strange to me. The authors might consider flipping the color scheme**

We will change the color scheme from having red for agreement in Figure 3. Thank you for this recommendation.

[Figure]

(a) This module helped me understand the topic addressed.

(b) My interest in watershed hydrology increased as a result of this lesson.

(c) This module excited me about a career in watershed hydrology.

(d) I would like to learn more in this area of watershed hydrology.

(e) My group worked together effectively in this module.

Percentage

S. Disagree
Disagree
Neutral
Agree
S. Agree

---

## Author Comment (AC2)

**Responses to Comments of Anonymous Reviewer #2**

1. **This manuscript deals with the important issue of how we best teach hydrology and particularly hydrological modelling. While the presented study certainly might have its value, I am afraid I could not really see this in the presented manuscript. There are no clearly formulated research questions and I found it hard to understand what actually had been done and why. So, after ready the text a few times, I feel more confused.**

Thank you for highlighting the need to clarify our introduction. We will revise the introduction to state the two primary research questions more clearly. We also modify the introduction in accordance with the specific comments of reviewer 3. The primary research questions are:

·        Is there consistency in results across the various methodologies for appraising the effectiveness of teaching modalities (i.e., student surveys, student assessments, in-class observations of student-instructor engagements)?
·        Do any of the three teaching modalities (i.e., lecture, student-led modeling, student-led design studio) lead to significantly improved student perceptions, student assessment scores, or the frequency of in-class engagements?

**Below are a few of the questions that I struggled with:**

2. **The results might be heavily influenced by the temporal sequence of the different ways of teaching. Is the study design with one course with a mixture of teaching approaches really suitable to study the differences of the different teaching approaches?**

We understand the reviewer's concern; however, we note that this experimental design is common. Other studies have shifted teaching modalities within a single course rather than conducting the experiment with separate groups of students. For example, many instructors studied the effect of shifts to online teaching as a result of COVID-19 pandemic (e.g. Khalil et al, 2020). Other non-COVID studies have similarly implemented the experimental design that we choose: several teaching modalities implemented in sequence (e.g. Maeng & Kim 2011; Limperos et al, 2015; Setyono 2016).

The primary advantage of this experimental design is that the core group of students is constant. As reviewer 2 mentioned in another comment, variations across groups of students can be large and may confound results. We are intentionally eliminating this variability by exposing the same group of students to different teaching approaches at different times. Though the temporal effect cannot be extracted from our study, we did choose to implement the modifications starting half-way through the course. Starting half-way through reduces the influence of students "warming up" to the instructor and course and gaining confidence as is expected during the first few weeks of any course.

We note that reviewers 1 and 3 requested that we include a discussion of uncertainty in this study. We will include a discussion of sequencing in this section.

COVID-19:

Khalil, R., Mansour, A. E., Fadda, W. A., Almisnid, K., Aldamegh, M., Al-Nafeesah, A., ... & Al-Wutayd, O. (2020). The sudden transition to synchronized online learning during the COVID-19 pandemic in Saudi Arabia: a qualitative study exploring medical students' perspectives. *BMC medical education*, *20*(1), 1-10.

Assessment of teaching changes within a single course:

Maeng, S., & Kim, C. J. (2011). Variations in science teaching modalities and students' pedagogic subject positioning through the discourse register and language code. *Science Education*, *95*(3), 431-457.

Limperos, A. M., Buckner, M. M., Kaufmann, R., & Frisby, B. N. (2015). Online teaching and technological affordances: An experimental investigation into the impact of modality and clarity on perceived and actual learning. *Computers & Education*, *83*, 1-9.

Setyono, B. (2016, January). Providing variations of learning modalities to scaffold pre-service EFL teachers in designing lesson plan. In *Proceeding of International Conference on Teacher Training and Education* (Vol. 1, No. 1).

3. **Were the authors also the teachers? From the text this seems so but I could not see this clearly stated.**

This detail cannot be disclosed per IRB protocols as it could potentially be used to identify the students who participated in the study. We note that most prior studies using TAR approaches (that were reviewed in our introduction) also did not disclose this information.

4. **The number of participants is low, does this allow drawing conclusions? We all know how variable student populations are and how much the general 'mood' can vary from year to year (often based on a few students who 'set the tone'**

We can interpret this comment as asking two different questions: 1) was the sample size large enough to produce statistically significant results or 2) was a sample of 20 students truly a representative sample of all hydrology students everywhere. We will attempt to answer both:

1) The concern is the possibility of false-negatives or false-positives that result from a population of 20 students. When discussing statistical significance, we always presented p-values for each test rather than just a binary significance report (i.e., the null hypothesis was rejected or failed to be rejected) with respect to an $\alpha$ threshold. The p-value is the probability that a significant result was observed when in fact there was no underlying mechanism, which numerically accounts for sample size. In most cases, results were sensitive at the $\alpha < 0.01$ threshold, which suggests that it was highly improbable that our results were just random chance due to small sample sizes. We

will add more to the discussion to explain results in the context of uncertainty where p-values are in the $0.01 < \alpha < 0.1$ range. Where results were not significant, we followed a similar approach. For example, in the case of paired differences in assessment grades we observed no significant difference. The concern in this case is that a low sample size could have potentially resulted in a false negative (i.e., there was a significant change in assessment scores across teaching methods, but we did not detect one because of a small sample size). Our reported p-value was not a borderline case that could easily change with a larger sample size.

For the response rates (Fig 5), we note that these regressions were not as limited by the class size as individual students could ask more than one question (or none), and the entire "experiment" was repeated across at least three periods for each teaching modality.  For the number of engaged students (Fig 6) we similarly were able to repeat this experiment across multiple class periods. We do note that some of these p-values are only sensitive at the $\alpha < 0.1$ and $\alpha < 0.05$ thresholds, which we do mention in Section 3.3, but will discuss in more detail in Section 4.

We agree strongly with the reviewer that  year to year variability occurs in this type of course, which was explicitly considered when designing the experiment. If conducted across multiple years, we hypothesize  that *cohort* would likely be a significant variable, and an unnecessary complication when attempting to analyze and discuss results. A result may have occurred not because of teaching methods, but as the reviewer says, simply because the "mood" differed. We decided that it was most informative to focus collecting more observations from one consistent group of students and one consistent instructor rather than across groups of students.

2) If the comment is asking if our group of students was truly representative of all hydrology students: this is more difficult to answer. The STEM education studies that we reviewed in our introduction had at most around 150 students. Even in these cases when sample size was larger, it is very unlikely that students all attending school in one geographic area were representative of all students everywhere. The students in this class were from a mixture of concentrations: civil engineering, environmental engineering, natural resources with a focus on water resources, and natural resources with a focus on forestry. I can only say qualitatively that there did not appear to be a subset of students who swayed the opinion of the entire class towards or away from any particular modality. We mention general school demographic data to give a sense of the environment that the students were in, as their responses may also be influenced by the more general academic environment in which they are participating. It is likely that our results are more relevant to students at similar institutions, and we will discuss this further in our discussion.

We note that reviewers 1 and 3 requested that we include a discussion of uncertainty in this study. We will include a discussion of sample size in this section.We will add the following to the discussion to address uncertainty of our results:

"4.4 Variability and Uncertainty of Results

Student responses to pedagogical practices may vary between cohorts of students and institutions. This study analyzed the effect of varying teaching modality on the same cohort of students to eliminate the effect of inter-annual variability of cohorts and instructors.  We also note that the general university/learning environment may influence the ways students engage with differing learning modalities and their responses to changes in modality.   We provide generalized background demographic data on this institution to provide the ability to compare between similar institutions. We anticipate our results to be more similar to student responses at institutions of similar size and composition.

Like many upper-level courses, this course was enrolled by a relatively small number of students, 24 students.  Due to the class size, we designed the study such that our engagement regressions were not limited by class size, as individuals could ask multiple questions.  Small class sample sizes are an obstacle that many upper-level course-based experiments may experience, but which is representative of the environment in which these topics are taught. "

5.  **What was the return rate of the questionnaires? How many accepted the link to the grades?**

The class had 25 total students. One student had to leave the course for personal reasons shortly after the midterm. A total of 20 students agreed to participate. The response rate was 20/24 = 83.3%. All students who agreed to be surveyed also agreed to have grades linked to survey results. We note that several students missed class when surveys were administered so the return rate for individual surveys was lower than the acceptance rate for research participation (18-lecture, 16-modeling, 18-design). The total number of surveys collected and average number of students in class (21.7 students/day on average) across the study period will be added to the manuscript.

6.  **The (very good) grades are of course highly influenced by the choice of questions and grading, the numbers alone do not say much**

We understand the reviewer's point, however, we first offer one clarification: none of our conclusions were based on the scores of any single assessment. The test variables were always the paired differences across tests. We will clarify that in the methods.  We understand that test difficulty will impact scores. We will make the individual assignments given to students available as part of the supplemental material.

Each assessment has been implemented and refined in this course over a number of years. The average score for each assessment in prior lecture-only offerings (which were not included in our IRB review) were all similar (i.e., no significant differences across assessments). As each individual assessment was approximately of the same difficulty, we believed that paired differences in assessment scores were an objective metric to capture significant improvement (or worsening). We observed no changes in grades, which mirrored previous years using lecture only.

**7. How many questions were there in the questionnaire? Only the five shown in figure 3? I am no expert, but I would assume there are better ways to design questionnaires to get more detailed information.**

It appears that possibly the reviewer did not see that the full questionnaire was included as a supplemental material. The reviewer's general point holds though: the survey was short. The survey design was kept short for two reasons:

1) A longer survey doesn't necessarily result in a better survey. The questions were chosen to align with the grand challenges in hydrology education that have been highlighted in prior research on this topic (Thompson et al. 2012; Ruddel & Wagner, 2015). We will make this point clearer in our revision. Our questions were aimed at understanding: 1) perceived value of the teaching method, 2) interest in the material, 3) interest in further learning, and 4) interest in career development. We included one additional question to control for potentially problematic group dynamics. The survey also included an open-ended feedback section to test for theme saturation. In this field, students could introduce new ideas possibly missed in the survey. Students primarily used this section to reinforce their responses to the previous likert questions. The results from the open feedback section suggested that we achieved theme saturation on the topic with only a few efficient questions.

2) We wanted students to report perceptions immediately after engaging with the material, and not some time later when the material and experience was no longer fresh. We left approximately 5 minutes of class time at the conclusion of each module for students to complete the assignments, which necessitated an efficient survey.

Ruddell, B. L., & Wagener, T. (2015). Grand challenges for hydrology education in the 21st century. *Journal of Hydrologic Engineering*, *20*(1), A4014001.

Thompson, S. E., Ngambeki, I., Troch, P. A., Sivapalan, M., & Evangelou, D. (2012). Incorporating student-centered approaches into catchment hydrology teaching: a review and synthesis. *Hydrology and Earth System Sciences*, *16*(9), 3263-3278.

---

## Author Comment (AC3)

**Responses to Comments of Reviewer #3**

**Summary: This is an interesting study that queries whether changing teaching modality impacts student-instructor interactions (really novel, very cool) and student learning. As an instructor, I really liked the emphasis of this study on interactions/questions – I think this is an interesting way to frame an analysis of this type. My only caveat to my comments is that I am not well versed in statistical analyses applied to educational assessments, so I am not able to comment on this part of the paper.**

**Major comments:**

1. **-I'd recommend revising and restructuring the introduction. The introduction has lots of good content, but felt a little disorganized, jumping around between general information and more specific hydrology information. There were also a few ambiguous statements in the introduction that I think could be sharpened (more in minor comments).**

The introduction will be revised. Specific changes are documented in the specific comment responses below (and in the responses to reviewers 1 and 2). We will add our research questions to the end of the introduction, as recommended by reviewer 1, and will significantly modify the introduction in accordance with other suggestions below.

2. **-A minor point, but I consider it major, given the focus of the study – the term 'student led learning' is introduced in the introduction, but not defined or explained. I think it's worth adding a few sentences to more clearly define this term and point to key references. This would broaden the introduction of this idea beyond only the 'flipped classroom' approach.**

Thank you for this comment.  We agree that the term should be more clearly defined and delineated from other, sometimes similarly-used terms.   We have added two references to help draw distinction between this term and others and revised the text as follows:

"There is broad evidence that student-led learning improves in-class experiences and retention of concepts in STEM education (de Jong, 2019). We define student-led learning to be a student-controlled learning process with continuous interaction and input from the instructor (Hoogenes et al., 2015), inclusive of both flipped classroom approaches and other project-based learning

paradigms. Student-led learning differs from other learning paradigms such as self-directed or self-regulated learning which tend to be unsupervised (Brydges et al., 2010).

3. **-I'd strongly recommend moving Table S3 to the main text – otherwise it isn't clear from the methods what questions are being used for assessment.**

Thank you for this recommendation. We will move Table S3 to the methods for ease of referencing all questions in one place.

4. **-While very interesting, my main concern is that this is only one year and one class of data. Thus, findings could be specific to those circumstances, and it is hard to say if this outcome would occur in another class and another year. However, I don't think that means this study should not be published. Instead, I'd encourage a thorough discussion of the limitations of this study in the discussion section.**

We will include a discussion on the choices we made for this experiment with regard to uncertainty and variability of results in the revised text, as detailed in our response to reviewer 2's comment #4.

**Minor comments:**

5. **Lines 39 – 40: "A recent series of interviews with water resources professionals indicated that graduates lacked critical workforce skills" – Could you add a little more information here? As written, I think this statement could lead to some confusion.**

The referenced study found that industry professionals thought incoming employees lacked knowledge on use and interpretation of data and different modeling systems. These details will be added to the introduction.

6. **Lines 49-50 aren't well integrated with the rest of the paragraph, which is about student-led learning – should these ideas come up later? Or could they be better connected to the rest of the paragraph? (Maybe move down to line 71?)**

Thank you for pointing out the lack of connectivity of this idea to the rest of the paragraph. We will move the reference to the following paragraph which similarly discusses data-driven analyses.

7. **Line 59: What is meant by 'more relevant material'? Could you be more specific here? Relevant to what and in what context?**

Thank you for this comment. Reviewer 1 similarly noted the ambiguousness of this phrasing. We will change the language to more clearly reflect the "career-relevance" of using teaching tools such as high-resolution datasets.

8. **Line 83: Another challenge in what respect?**

In this paragraph we intend to highlight a lack of research using individual-level data in contrast to the wide availability of studies that used aggregate-level perception data. Upon review, we agree that using the phrasing 'another challenge' alone, does not convey this current research gap. We will revise the phrasing to more clearly demonstrate this gap as follows:

"Prior studies place strong emphasis on group mean outcomes of hydrology courses (i.e., average perceptions, average assessment scores) rather than the outcomes of individuals (Gallagher et al., 2021; Knoben & Spieler, 2022; S. W. Lyon et al., 2013; Merck et al., 2021; Pérez-Sánchez et al., 2022), a current gap in hydrology education research."

9. **Line 100: 3000-level is institution specific – (my institution uses 300 level, for instance) – could you use another way to contextualize the course level that translates across institutions? Maybe just refer to this as 'upper level'?**

Reviewer 1 similarly pointed out this terminology challenge. We will adjust the language to represent, as recommended by reviewers 1 and 3 to be 'upper-level' and include expected courses previously taken.

10. **Line 125: 'on campus' might be too colloquial – maybe 'local'?**

We will adjust the language as recommended.

11. **Line 243: worth looking at interquartile ranges? Did you bring the lower grades up with the shift in teaching modality?**

We used two-sample Kolmogorov-Smirnov (KS) tests to compare the distributions. This statistical approach tests for any changes in the distribution (including shifts in the extremes), not just the means or medians. None of the KS-tests showed a significant difference which suggested that there was no significant change in the lower grades.

12. **Line 293: should this be 'reported'?**

Thank you for catching this error. Yes, we will correct it as recommended.

13. **Line 294 – 296: I like this conclusion**

Thank you for this positive feedback!

**14. Line 298: have other studies done this? Is this an approach that is used in the educational literature?**

In our experience (and review of the literature) it is common for instructors to track written assessments scores at the individual level, but only the average of student perceptions (e.g., mid- and end of-term student evaluations of teaching). To our knowledge, other studies have not presented an individual-level analysis of student perceptions in hydrology education, and this is a primary result of this study. The likert results (Fig. 3) and alluvial diagram (Fig. 4) together show that the mean outcome might remain constant, but the impact on individual students can be large (similar to written assessments, Fig. 7). We are recommending that this approach should be more widely adopted particularly because the changes in assessment scores did not match reported changes in perceptions (Fig. 7). We will clarify this point in the revised text.

**15. Line 310: Yes – I think this is possible. I've seen students doing group work divide and conquer on assignments, meaning that they may miss out on learning because they have self-selected to do a portion of the assignment that doesn't involve "x" activity.**

We have similarly observed the "divide and conquer" approach. One group in particular was skilled in how they split up tasks. Although their final work was exceptional, each student seemed to only master one particular task. Given our IRB protocols, it is not possible to link this group to specific responses or grades as all results were anonymized. We were also not able to find any published studies that specifically looked into this effect. In contrast, most studies promote group learning as a panacea and do not consider the possibility for individual learning to lag that of the group. We are proposing to leave this wording as is so that we don't overemphasize a point for which we don't have supporting literature or direct observations from our study aside from a minor addition to point to the "divide and concur" strategy : "...lowered student engagement *in individual assessment components."*

**16. Line 325: References got a little messed up here!**

Yes, this was auto-generated by Zotero. I've checked the reference database and don't see any obvious problems. We will go through the manuscript and manually correct issues with the references.

**17. Figure 2: Could you update the legend to have spaces and be written text, not abbreviations?**

Thank you for pointing this out.  We will modify both figures 1 and 2 in accordance with this comment, and that of Reviewer 1. See modified figure with responses to Reviewer 1 comments.

**18. Figure 3: Is it possible that interest in a career in hydrology merely increased through time, and not as a result of a particular approach to teaching? I don't think your study design allows you to separate temporal effects (if my assumption that the delivery timeline was lecture -> modeling -> design project), so it may be worth pointing this out (but my assumption may be incorrect).**

Yes, this is possible; however, we do note that these teaching modalities were introduced in the second half of the course. Students already had 8 weeks of exposure to hydrology concepts and techniques to this point. We will include in our discussion of uncertainties the role of sequencing the teaching modalities as follows:

"These modality modifications were implemented in-sequence over the last half of the semester. This design was chosen such that students would already be comfortable with the hydrological concepts, instructor, and course environment and eliminate the 'warm-up' period to the new course environment. However, this study design does not allow for separation of the in-sequence influence of time from the results. We found that aggregate interest in careers in hydrology did not significantly increase with time, though individual students changed their perspectives with each modality, leading us to the conclusion that time was not a significant variable in these results."

**19. Figure 4: I love a good figure, but I struggled to see what the authors wanted me to see in this figure. Would there be some way to highlight a key message, or include a number or a few numbers with each graph, or even a summary of the key takeaway message in the caption?**

As per recommendations by reviewer 1, we have modified Fig. 4 to include all modality comparisons (i.e. 3 columns instead of 2 columns of plots). The key conclusion here lies within the tracking of students who shifted their perceptions of the course dramatically between modalities. Despite average aggregate level conclusions that the three modalities were equally perceived (Fig. 3) There are certain students in each modality who dramatically shift their perceptions with modality. We will add detail to the results section and the figure caption to ensure that detail is not overlooked. We believe that together Fig. 3 and Fig. 4 demonstrate the necessity to have both aggregate-level and individual-level data analysis.

**20. Figure 7: Could be moved to supporting information – I found this the least interesting! It was challenging to see anything in this figure. I also think this figure could be redesigned if you wanted to include it in the paper. For instance, add the question text above each section (so readers aren't flipping back and forth between different parts of the manuscript), and add significance level from the statistical test to each figure.**

Thank you for this suggestion. We will move this figure to the supplemental material and modify the figure such that the questions are visible on the figure itself as displayed below:

[Figure]

**Supplemental Figure:** Cumulative distribution functions of student grades for students who responded to the career and interest questions with a '4' or '5' on the Likert scale of '1'-'5'. Questions refer to the Likert survey questions to which each response corresponds. Column A compares Lecture and Model modalities, column B compares Lecture and Design modalities, and column C compares Model and Design Modalities.